# When Your Child Has Cancer: A Path-Analysis Model to Show the Relationships between Flourishing and Health in Parents of Children with Cancer

**DOI:** 10.3390/ijerph182312587

**Published:** 2021-11-29

**Authors:** Carmen Pozo Muñoz, Blanca Bretones Nieto, María Ángeles Vázquez López

**Affiliations:** 1Department of Psychology, University of Almeria, 04120 Almeria, Spain; cpozo@ual.es; 2Maternity and Child Hospital, Torrecárdenas University Hospital, 04009 Almeria, Spain; mavazquezl59@gmail.com

**Keywords:** childhood cancer, flourishing, health, wellbeing, distress, coping, social support, path analysis

## Abstract

Background: Childhood cancer is a disease with a psychosocial impact on parents who experience health problems and distress. Their reactions depend on the relationship of multiple factors. The objective of this paper is to evaluate the interrelationships between flourishing and the variables linked to the health and wellbeing of parents of children with cancer. Methods: Mothers/fathers of children with cancer participated in an exploratory study in response to a series of questionnaires. Likert-type scales were used to measure perceived health, wellbeing, flourishing, stress, coping, and social support. Results: Out of a total of 94 children, 138 parents (60 men/78 women) are represented. Participants show physical symptoms and an unstable coping pattern. A path analysis model is presented. As to the goodness of adjustment of the statistics used, good results were obtained. Flourishing tends to coexist with wellbeing, while flourishing coexists negatively with symptoms. There is an indirect relationship between flourishing and poor health. There is a positive relationship between flourishing and coping, as well as between flourishing and satisfaction with the support received (especially from sons/daughters). This support was negatively related to the subjective health report. Conclusions: Flourishing is shown as a healthy coping strategy. The results can enrich the development of psychosocial interventions aimed at promoting adequate adaptation.

## 1. Introduction

In Spain, a total of 31,073 cases have been registered between 1980 and 2020 among the population aged 0–14 years, according to the latest report from the Spanish Registry of Childhood Tumors and the Spanish Society of Pediatric Hematology and Oncology—RETI-SEHOP [1]. Childhood cancer is a social problem and a vital stressful event, not only for the diagnosed child but also for the parents, affecting their physical and psychological health. Pediatric cancer is considered an unpredictable and uncontrollable stressor that affects the family as a whole and parents, in particular, in their role of caregiver [2,3]. Specifically, it has been shown that parents of children with cancer have a higher prevalence of anxiety, depression, and Posttraumatic Stress Disorder (PTSD) compared with population controls [4].

Individual, intra-family, and contextual factors are important in explaining family adjustment perceived by parents who face a diagnosis of cancer in their children [5]. Specifically, what is agreed is that parents see their quality of life diminished and have to encounter multiple sources of stress, such as waiting for medical results, uncertainty or helplessness when receiving the prognosis, and witnessing the child’s pain, etc. Moreover, another factor should be taken into account, parents suffer symptoms related to their children’s health problems many years after the end of cancer treatment [6,7].

Outside of the hospital context, the diagnosis and treatment of childhood cancer disrupt family functioning in different ways: Family members present numerous challenges in daily routines, social relationships, and at work (loss of earnings, reduced working hours or being on sick leave). Furthermore, economic difficulties arise prompted by unforeseen expenses, such as visits to other hospitals at a greater distance from where they live [8]. In this sense, the parents commonly use problem-focused (e.g., seeking help from professionals and support groups) and emotion focused (e.g., behavioral distractions, venting, and crying) strategies to solve day-to-day conflicts arising from their child’s illness [9]. However, the studies agree that the strategies implemented by parents of children with cancer are not truly adaptive. Studies have shown a correlation between coping and somatic symptoms. In addition, many parents use emotion-focused coping strategies rather than problem-focused coping strategies [10,11].

At the same time, after their child’s diagnosis, parents commonly require the support of family members, including their parents and siblings [12]. It has been shown that the perception of social support provides health and wellbeing benefits [13]. In addition, its role has been studied by the parents of paediatric patients, reinforcing its usefulness as an adaptive coping strategy, which is required in different ways during the stages of the disease [14]. Social support is a fundamental element for parental wellbeing, coming both from the formal support network (health personnel, bosses, etc.) and the informal network, such as one’s partner and the rest of the family [15].

Aside from the coping strategies that are traditionally studied, some studies from Positive Psychology have recently pointed to other abilities or strengths that make individuals more effective in coping with highly stressful situations. Furthermore, it has been proven that positive psychology interventions can be advantageous to improve the interaction between parents and children in daily life. One of these is “flourishing” [16].

Different authors have defined this construct in various ways. For instance, Reference [17] refers to Flourishing as a combination of central features (positive emotions, engagement, sense of meaning, and purpose) and additional features (self-esteem, optimism, resilience, vitality, self-determination, and positive interpersonal relationships). Huppert and So [18] defined it as a mixture of feeling good and functioning effectively, having better social relationships, demonstrating a positive coping attitude towards stressful events, and experiencing fewer obstacles in daily activities. 

Previous research revealed that flourishing individuals more often use adaptive coping strategies and function positively in all aspects of life [19]. In addition, individuals who have high flourishing are likely to have social competence which, in turn, contributes to diminishing the susceptibility to psychopathology and enhances resiliency [20]. In sum, the difference between Flourishers or non-Flourishers is the fact that the first are expected to have excellent mental and physical health and are more resilient to vulnerabilities and challenges [21,22].

The role of flourishing has also been studied in other contexts, for example, in education. Despite the fact that some authors affirm that limited research has been carried out to assess the role of flourishing in the educational area [23], some studies show that students who have this attitude act to achieve their goals with greater confidence and competence, acquire self-control skills, experience positive emotions, and show increased social and moral awareness [24]. Flourishing has also been incorporated into the occupational health framework as a beneficial factor for workers [25], and has been researched as a predictor for recovery from mental disorders [26]. More precisely, improving flourishing mental health seems particularly relevant for anxiety disorder patients. 

In the health area, it has been shown that the use of some coping strategies has positive consequences on wellbeing [27], improving the prevention of diseases and facilitating the adaptation of the primary caregiver to the circumstances of their family member [28]. In the same way, the existence of an association between flourishing and experimenting with fewer limitations in daily activities as well as enjoying good health has been verified [18,29,30]. Some researches [31] show relevant implications of flourishing for psychosocial or clinical interventions that improve subjective wellbeing through the modification of levels of flourishing.

Based on this perspective, this work looks at flourishing as a potential coping strategy (a first in this field of research), together with the other classically considered strategies, in order to analyze its role in the health and wellbeing of parents of children with cancer. 

Specifically, we aim to explore whether in a stressful situation of cancer diagnoses of a child the levels of flourishing are related to coping strategies and the physical and psychological wellbeing of the parents. After regressing the levels and frequency of stress, as well as the treatment status, on all the variables (to control their effect on the rest of the variables), we try to determine the amount of variance that flourishing can share with the variables related to coping strategies, wellbeing, and the physical and psychological state of health. In addition, the effect of the coping strategies on wellbeing and the physical and psychological state of health will also be taken into account to determine the indirect relationships between flourishing and these latter variables. In short, as a result of what has been argued above, a negative relationship is expected between flourishing and suffering from symptoms, as well as between flourishing and a negative self-assessment of health status. However, we predict that the correlation between flourishing and wellbeing will be positive.

On the other hand, there will be a greater correspondence between active coping strategies (for example, “positive reappraisal” and “problem-solving coping”) and flourishing. It is understood that to “flourish” it is necessary to endure and learn from experiences (far from being avoided) as a way for growth and proper development. It is a “state of being” in the face of adversity [32]. Likewise, the perception of the support received by the family will be more positively related to flourishing than to the other sources. This is accentuated in this context of illness, where the closest family plays an important role in facilitating care [33].

## 2. Materials and Methods

### 2.1. Study Population

This study was conducted within the framework of the *Psychosocial Repercussions of Childhood Cancer in Parents*, with a focus on investigating the main factors involved in the health and wellbeing of parents with children affected by cancer.

The participants were mothers and fathers of children diagnosed with cancer and treated in a hospital in southern Spain. In this hospital, between 15 and 20 cases are diagnosed on average per year. A total of 181 cases were diagnosed and the parents of 131 children were accessed, which represents obtaining information related to 72.38% of cases. Since our sample is made up of the parents of these children, a percentage of 38.12% was reached.

In terms of recruitment rates, the exclusion criteria were as follows: Parents whose children died as a result of the disease, 30 in this case, and 20 were excluded due to the language barrier (given that the incorrect understanding of the survey’s content might lead to bias). During the period that the questionnaires were conducted, we “lost track” of 12 cases (due to families travelling to hospitals in other cities at the time the information was collected), while the parents of 26 patients refused to participate. Furthermore, in 18 cases, only the mothers attended the interview. Finally, in 4.3% of the total (six cases), the parent was separated from his/her partner and did not provide any contact information. A final total of 138 parents participated in the study.

### 2.2. Data Collection

The study was approved by the Research Ethics Committee of the Almeria Center of the "Torrecárdenas" University Hospital, and was conducted in accordance with the Helsinki Declaration of 1975 (as revised in 2013) [34] with respect to the participants and the confidentiality of the collected data. The participants were all required to sign a statement of informed consent. Trained interviewers collected the information.

### 2.3. Measures

The participants were all interviewed with the application of a set of questionnaires. During the application, the participants were asked to contextualize their answers according to the specific disease situation of their children. These were administered individually (lasting about 1 h), and included the following variables: 

#### 2.3.1. Demographic and Clinical Characteristics

The interview included the demographic and clinical-related characteristics of the participants: Gender, age, marital status, nationality, level of education, occupation, child’s diagnosis, date of diagnosis, type of treatment, and the current status of treatment.

#### 2.3.2. Physical and Psychological State of Health 

An adaptation of the Symptom Scale [35], based on the original version by Jou and Fukada [36], was used to assess the presence of physical and psychological symptoms. The scale contained nine items. The score ranged from 1 to 5, with higher scores indicating worse health status. The parents’ opinions regarding their own state of health was gathered via an ad hoc item. This item is “*in general, I consider myself to be in very bad health*” (subjective health). The response options were the same as those previously described, but the higher score indicates better subjective health.

#### 2.3.3. Wellbeing

The Spanish version [37] of the Satisfaction with Life Scale [38] was used to understand the level of wellbeing experienced by the parents. This measure is a 5-item self-report. Parents responded to each item in a 5-point response scale, with a higher score indicating greater life satisfaction. 

#### 2.3.4. Stress in Parents

A reduced version of the Paediatric Inventory for Parents [39] was used. This scale consists of 15 items, grouped into four subscales: (1) Communication, (2) emotional functioning, (3) medical care, and (4) role function. The frequency of exposure to a stressful situation related to this childhood disease was studied (using a Likert-type response format, with “1” signifying “*never*” and “5” signifying “*very often*”). Then, participants were asked about the psychological impact caused by those circumstances (“1” signifying “*nothing at all*” and “5” signifying “*very much*”). 

#### 2.3.5. Coping with Stress 

We assessed the frequency at which seven basic coping strategies were implemented (21 items): Problem-solving coping, negative auto-focused coping, positive reappraisal, overt emotional expression, avoidance coping, social-support seeking, and religious coping—considering responses ranging from “0”, “*never*” to “5”, “*almost always*”. These strategies belong to the reduced version of González and Landero’s Coping Stress Questionnaire (CAE) [40]. As we administered the scale, the word “problem” was replaced by “disease” for a better adjustment. 

#### 2.3.6. Perceived Social Support 

The parents’ assessment of their formal networks (healthcare professionals) and informal networks (family, friends, neighbors, and workmates) was obtained by completing the adapted version of the Perceived Social Support Scale (EASP/PSSS) [41] by the authors of [15]. The sources and types of support were evaluated, as well as the parents’ satisfaction with this support (whose results are used in this article). This last variable was measured using a 5-point response scale (from “*very dissatisfied*” to “*very satisfied*”). 

#### 2.3.7. Flourishing

Flourishing is incorporated as another coping strategy using the Spanish version of the Flourishing Scale [42], which consists of eight items, each measuring a core aspect of optimal psychological-functioning on a 5-point Likert scale (“disagree-agree”). “*I lead a purposeful and meaningful life*”, “*I am competent and capable in the activities that are important to me*”, and “*I am optimistic about my future*” are examples of items on the Flourishing Scale. 

The CAE and the Flourishing Scale are validated and internationally used instruments.

### 2.4. Statistical Analysis

First, descriptive analyses (mean, standard deviation, and correlations) for continuous variables and frequency distribution analyses for categorical variables were performed.

Next, to evaluate the inter-relationships among the study variables, we used the path analyses. Due to the limited sample size, we performed the analyses in different stages. First, in the separated analyses, flourishing was regressed on wellbeing, objective health, and subjective health, on all of the coping and social support variables, as well as on the treatment status (“*current situation*”), stress impact, and stress frequency (to control their influence on the relationship between flourishing and the rest of variables). Subsequently, after the non-significant regression coefficients were fixed to 0, we ran a model with all of the mentioned variables together. However, this time, flourishing was regressed on all of the coping and social support variables. The non-statistically significant relationships were all fixed to 0, except those whose change would lead to a worsening of the model fit.

The full information maximum likelihood method was used to estimate the missing values. The model fit was checked using the Comparative Fit Index (CFI) and the Root Mean Square Error of Approximation (RMSEA) with a 90% confidence interval. The CFI values greater than 0.97 and RMSEA values less than 0.05 are indicators of good model fit [43]. The analyses were all carried out using the MPlus v6.0 program (Muthén & Muthén, Los Angeles, CA, USA) [44].

## 3. Results

Table 1 presents the descriptive statistics of the sociodemographic characteristics for the participants and the clinical condition of their children. Most of the participants were Spanish (84%), aged from 41 to 50, married or living together as a couple, educated to the high school level, with almost half working in the service sector (including subsectors, such as commerce, communications, call centers, finance, and tourism, etc.). Their children were diagnosed with leukaemias (*n* = 61; 44.2%) and solid tumors (77; 55.8%. Specifically, central nervous system tumors (*n* = 19; 13.77%), lymphomas (*n* = 15; 10.87%), bone tumors (*n* = 12; 8.7%), Wilms’ tumor (*n* = 12; 8.7%), neuroblastomas (*n* = 7; 5.07%), germ tumors (*n* = 6; 4.35%), retinoblastoma (*n* = 1; 0.72%), and others (*n* = 5; 3.62%). The most common treatment was chemotherapy and the majority of the children had been receiving treatment until only a few years before.

First, the results obtained from the descriptive analyses of the studied variables are provided in Table 2. These allow us to understand the psychosocial variables related to childhood cancer on the parents of the affected children. The participants do not experience serious health problems in general, but they do manifest certain symptoms related to the pathology of their children (the mean score of the Symptom Scale does not exceed the midpoint). Regarding the answers on “subjective health”, it is observed that the opinions are not very unfavorable. In parallel, the parent’s perspective on “perceived wellbeing” is positive.

On the other hand, the frequency of exposure to stressors as a result of their child’s illness is relevant, along with the associated emotional impact. These, in turn, present a pattern of heterogeneous coping. The highest results are observed for “problem-solving coping” and for “negative auto-focused coping”, respectively. The satisfaction-related variables with support coming from the formal and informal support networks obtained a score of more than 4/5. Parents demonstrated a flourishing attitude, which is understood as a coping strategy. In addition, the results of all the correlations between the variables studied are included.

Second, the individual associations between variables (partial correlations) are presented in Figure 1, whereas the correlations omitted in the figure are displayed in Table 3. 

The proportion of explained variance for wellbeing, objective health, and subjective health was 0.32, 0.12, and 0.20, respectively.

In relation to the model’s goodness-of-fit statistics, Chi-squared, X2 (df) = 190.789 (192); *p* = 0.5111. The CFI and TLI were equal to 1, whereas the RMSEA (IC 90%) was 0 (0.004). 

The results of these analyses allow us to report two direct relationships: Flourishing—wellbeing and flourishing—objective health. The directions of the relationships indicated that a greater presence of flourishing strategies tended to co-occur with a greater presence of wellbeing, while flourishing co-exists negatively with experiencing symptoms. Furthermore, the indirect relationship between flourishing and subjective bad health could lead to a similar interpretation as the direct relationship between flourishing and objective health. Therefore, the levels of flourishing strategies were positively associated with the satisfaction support with sons and daughters. However, this last variable was negatively linked with the report of subjective health. In other words, the perception of health is more positive.

Altogether, these results point to two main directions: (1) The importance of the flourishing strategies when people report symptoms associated with the circumstance of their child’s illness and report suffering poor health as a result, and (2) the direct relationship of flourishing and wellbeing in a stressful situation of cancer diagnoses of a child. Furthermore, it is important to highlight that these relationships have been controlling the effects of the stress impact, stress frequency, and the treatment situation.

Finally, the model identifies a positive relationship between flourishing and certain forms of coping (positive reappraisal and problem-solving coping), as well as between flourishing and satisfaction with the support received from some sources (friends, co-workers, siblings, children, and neighbors).

## 4. Discussion

First, the results obtained indicate that the parents who participated in this study can be considered as suffering from certain physical symptoms. They experience stress, reporting that they feel affected by it. Despite this, they do not have a catastrophic perception of their own health and enjoy acceptable levels of wellbeing. These findings relate to their role as primary caregivers, manifesting “active cognitive coping,” which involve developing a determination to confront the existing difficulties, suppressing their own needs [45,46,47].

With regard to the coping pattern, there was no definite style. A focus on problem solving and negative auto-focus are the most used forms. On the other hand, the participants have a flourishing attitude and, in terms of social support, receive this from their formal and informal networks, and are satisfied with it. 

Second, the most important achievement of this paper is the demonstration of a model, which confirms that flourishing is an important promoter variable of health and wellbeing regarding the studied population group. 

In this case, flourishing relates to the role of parents as caregivers. In addition, it is linked with the description of Huppert and So [18], as a predisposition towards “engagement”, the search for a behavior with a “sense” or meaning, oriented towards a “purpose” in itself. Moreover, the feeling of “competence” is positively related to perceiving oneself as responsible for the happiness of someone you love. This circumstance can also explain and enhance the power of flourishing, which is positively related to satisfaction with child support, and the latter which is negatively related to subjective health. Likewise, there is a parallel with the coping style, oriented towards the meaning proposed by Folkman [48]. Therefore, parents with a flourishing attitude have the ability to positively channel a stressful event, redirecting it to a meaning, and an ultimate purpose. According to the study by Lazarus [49], flourishing would act in a way to make the result of an initially “unfavorable” stressful event (according to the stress and coping model), positively influence the parents’ assessment of what they can do to change a situation, unsettling them or provoking an emotional imbalance. In this way, the results obtained would be justified. In other words, the coexistence of a positive relation between flourishing and “active” coping strategies, specifically, positive reappraisal and problem-solving coping.

In addition, the model shows that people who score high in flourishing are more satisfied with various sources of social support (friends, coworkers, neighbors, and children). In fact, the happiest people enjoy good interpersonal relationships [50]. In contrast, we tend to feel sad when we are lonely and cheerful when we are with other people. This happens to people of all ages and cultures [51]. Moreover, as has been traditionally observed in other studies, there is a positive connection between satisfaction with these support sources (in this case from children, couple, and grandparents), with enjoying good health, and wellbeing [10,11,12,13,14,15]. All of the above represent a model in which relations act by controlling the frequency and impact of stress, as well as the current situation regarding the child’s treatment.

## 5. Conclusions

Certainly, the information provided in this research is scientifically relevant, taking into account its objectives. This article confirms, in a novel way, the effect of flourishing as an effective and healthy coping strategy on the parents of children with cancer.

One of the aspects that should be highlighted is related to the number of participants that make up the sample. Here, it must be considered that the study was carried out in a small city, but it was applied to residents of the capital and province. In addition, it was very difficult to recruit the parents who finally participated, due to the sensitivity of the subject. This may be associated with the fact that, for most of the parents, submitting to an interview was a great emotional effort, in which they had to exercise awareness and delve into circumstances of their lives that were especially painful for the child, the parents themselves, and the rest of the family. At the same time, the language barrier (in the immigrant population) and the families who turned down the invitation to participate were further difficulties to contend with.

Furthermore, the child’s situation regarding the treatment and care they needed (including “hospital leave” periods, between chemotherapy cycles, which they spent at home) as well as the parents’ difficulty in combining this circumstance with the rest of their daily responsibilities, was reflected when trying to make an appointment for the session. In this regard, “childhood cancer” research had already been found to be a difficult field in which to accumulate large samples [52].

Regarding clinical implications, flourishing can be translated into a “form of action” or behavior. Therefore, it would be convenient to take these results into account in order to develop an intervention program with a quasi-experimental design, aimed at training in flourishing for the parents of children with cancer.

## Figures and Tables

**Figure 1 ijerph-18-12587-f001:**
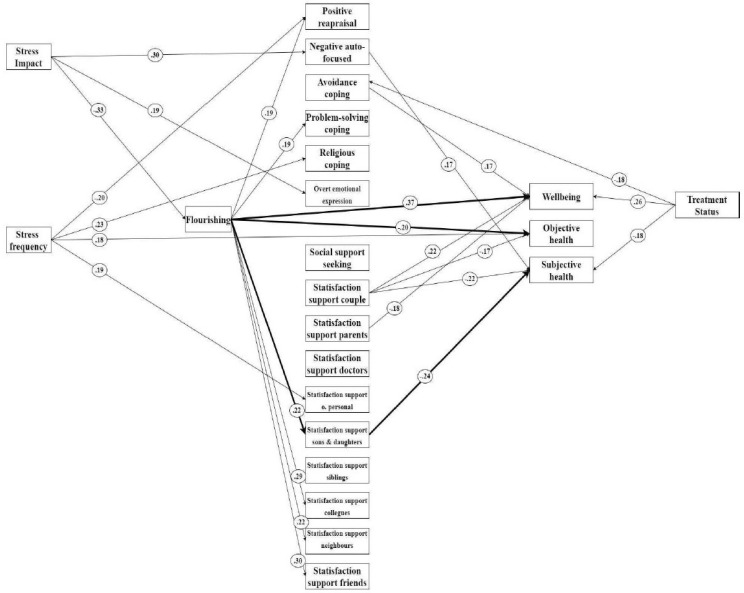
Path analysis model: Flourishing, social support, and coping, and their relationship with wellbeing and health.

**Table 1 ijerph-18-12587-t001:** Sociodemographic characteristics of parents and clinical conditions of their children.

	Sample (%)		Sample (%)
**Age**		**Occupation**	
21–30	6	Unemployed	21.7
31–40	36	Primary sector	13.7
41–50	44	Secondary sector	10.7
51–60	14	Third sector	53.9
**Nationality**		**Diagnoses in Children**	
Spanish	84	Leukemia	44.2
Others	16	Solid tumors	55.8
**Marital Status**		**Type of Treatment**	
Spouse/partner	88.5	Chemotherapy	67
Divorced	11.5	Surgery	12
**Education**		Surgery and radiotherapy	4
No qualifications	5	The three types	17
Primary	34.8	**Current Situation** **“treatment status”**	
Secondary	40.6	Receiving treatment	28.2
Graduates	19.6	Out of treatment ≤ 5 years	34.1
		Out of treatment > 5 years	37.7

**Table 2 ijerph-18-12587-t002:** Screening on the psychosocial functioning in parents of children with cancer: Correlation between the analyzed variables.

Variables	*n*	Mean	SD	1	2	3	4.1	4.2	5.1	5.2	5.3	5.4	5.5	5.6	5.7	6	7	7.1	7.2	7.3	7.4	7.5
Symptoms (objective health) (1)	138	2.29	0.96	-																		
Subjective health (2)	138	2.18	1.43	0.473 **	-																	
Wellbeing (3)	138	3.16	1.15	−0.238 **	−0.191 *	-																
Stress Frequency (4.1.)	138	3.91	0.61	0.213 *	0.057	0.175 *	-															
Impact caused by stress (4.2.)	138	2.82	0.65	0.219 **	0.063	0.109	0.541 **	-														
Problem-solving coping (5.1.)	138	2.39	1.15	−0.101	−0.091	0.158	−0.120	−0.051	-													
Positive reappraisal (5.2.)	138	1.80	1.25	−0.246 **	−0.206 *	0.265 **	−0.221 **	−0.086	0.010	-												
Social support seeking (5.3.)	138	1.56	1.25	−0.009	0.008	0.105	0.084	0.096	−0.088	0.164	-											
Religious coping (5.4.)	138	1.28	1.49	0.061	−0.076	−0.062	0.236 **	0.145	−0.023	−0.094	0.294 **	-										
Avoidance coping (5.5.)	138	1.24	1.07	0.004	−0.024	0.202 *	0.041	0.059	−0.050	0.236 **	−0.088	−0.092	-									
Overt emotional expression (5.6.)	138	0.95	0.91	0.167	0.156	−0.057	0.223 **	0.211 *	−0.008	−0.043	0.079	0.039	0.036	-								
Negative auto-focused coping (5.7.)	138	2.06	0.90	0.185 *	0.245 **	−0.188 *	0.246 **	0.310 **	−0.161	−0.328 **	0.064	0.139	0.081	0.247 **	-							
Flourishing (6)	138	3.85	0.63	−0.282 **	−0.198 *	0.403 **	−0.116	−0.323 **	0.186 *	0.211*	0.110	0.029	0.111	−0.171 *	−0.248**	-						
Satisfaction with perceived social support (7)	138	4.26	0.71	−0.068	−0.126	0.099	−0.052	−0.106	0.010	0.137	0.240 **	0.173 *	0.040	−0.111	−0.029	0.293 **	-					
Children (7.1.)	138	4.92	1.27	−0.164	−0.251 *	0.074	−0.257 *	−0.287 **	0.193	0.214*	0.185	−0.036	0.015	0.055	−0.121	0.359 **	0.522 **	-				
Nurses and nursing assistants (7.2.)	138	4.59	0.88	0.045	0.006	0.029	0.125	0.047	0.060	0.086	0.048	0.077	0.093	0.089	0.072	0.151	0.563 **	0.202	-			
Parents (7.3.)	138	4.45	1.41	0.051	0.062	−0.196 *	−0.047	−0.053	−0.025	−0.081	−0.020	0.190 *	−0.194 *	0.035	0.096	−0.047	0.520 **	0.140	0.206 *	-		
Doctors (7.4.)	138	4.42	1.00	−0.112	0.077	0.067	−0.047	−0.023	0.093	0.085	0.123	−0.021	0.083	−0.067	0.032	0.157	0.502 **	0.088	0.654 **	0.215 *	-	
Siblings (7.5.)	138	4.33	1.24	−0.039	0.016	0.000	−0.040	−0.143	−0.046	−0.044	0.104	0.107	−0.027	−0.038	−0.036	0.126	0.528 **	0.236*	0.183*	0.480 **	0.122	-
Couple (7.6.)	138	4.26	1.27	−0.232 **	−0.324 **	0.283 **	−0.125	0.033	0.087	0.248 **	0.154	0.093	−0.073	−0.210 *	−0.071	0.164	0.426 **	0.389 **	0.125	0.059	0.160	−0.006

* *p* ≤ 0.05; ** *p* ≤ 0.01.

**Table 3 ijerph-18-12587-t003:** Correlation coefficients omitted in Figure 1.

Correlations	Stress Impact	1	2	3	4	5	6	7	8	9	10	11	12	13	14	15	16	17	18	19
Stress Frequency (1)	0.52 **																			
Positive reapraisal (2)	-	-																		
Negative auto-focused (3)	-	-	−0.31 **																	
Avoidance coping (4)	-		0.27 **	-																
Problem-solving coping (5)	-	-	-	-	-															
Religious coping (6)	-	-	-	-	-	-														
Overt emotional expresión (7)	-			0.16 *	-	-	-													
Social support seeking (8)	-		0.19 **		-	-	0.29 **	-												
Statisfaction support couple (9)	-		0.16 *	-	0.13	-	0.12	−0.25 **	0.17 *											
Statisfaction support parents (10)	-	-	-	-	−0.18 *				−0.11	-										
Statisfaction support doctors (11)	-	-	-	-	-	-	-	-	0.13	-	-									
Statisfaction support o. personal (12)	-	-	-	-	-	-	-	-	-	-	-	0.66 **								
Statisfaction support sons & daughters (13)	-	-	-	-	-	-	-	-	-	0.33 **	-	−0.19 *	0.20 *							
Statisfaction support siblings (14)	-							-	-	-	0.47 **	-	0.10	0.22 **						
Statisfaction support collegues (15)	-	-	−0.14	-	−0.19 *	0.17 *	0.13	-	-	-	0.26 **	-		-	-					
Statisfaction support neighbours (16)	-	-	-	-				-	-	-	0.24 **	-	-	-	-	0.62 **				
Statisfaction support friends (17)	-	-	-	0.17 **			−0.09	-	0.18 **	0.12 *		-	-	-	-	0.40 **	0.64 **			
Objective health (18)	-	-	-	-		-	-	-		-		-	-	-	-	-	-	-		
Subjective health (19)	-	-	-	-	-	-	-	-	-	-	-	-	-	-	-	-	-	-	0.45 **	
Wellbeing (20)	-	-	-	-	-	-	-	-	-	-	-	-	-	-	-	-	-	-	−0.07	−0.01

* *p* < 0.05 ** *p* < 0.001.

## Data Availability

Not applicable.

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
