# Peer review of "When Your Child Has Cancer: A Path-Analysis Model to Show the Relationships between Flourishing and Health in Parents of Children with Cancer"

_ijerph, 2021, doi:10.3390/ijerph182312587_

Round 1

Reviewer 1 Report

This is an exploratory study on the associations of flourishing with perceived health and wellbeing of the parents of children with cancer. In the introductoin, the authors state that aside from the coping strategies that are traditionally studied, some studies from Positive Psychology have recently pointed to other abilities or strengths that make individuals more effective in coping with highly stressful situations. They say that positive psychology interventions can be advantageous to improve the interaction between parents and children in daily life, and one of these positive consepts is "flourishing". 

This is an interesting point of view, but the authors do not describe more thoroughly whether this coping mechanism/attitude could be learned/practized or is it purely an individual "genetic" mindsetting.

In the beginning of the introduction, the authors state "The latest records indicate that childhood cancer incidence has increased considerably over recent decades". However, the references are not fully supporting this interpretation. The reference nb 1 describes the incidence in chidren below the age of 5 years at diagnosis, and ref nb 2 describes prediction model of the incidence (on the list, the name of the publication, journal and year are missing). So,  I think that it is not fully correct to say that "childhood cancer incidence has increased considerably" even though a trend is most evident. Still most unclear whether improved registration and dignostics play a major role in this trend or not.

In "Materials and methods" the authors should describe when the data was collected, what is the annual number of new cases at that hospital, how they ended up stating that response rate was 50%.

It is said that interview was performed but then questionnaires are mentioned, to. Was it so that the data is based on validated questionnaires? Specifically, it would be good to know whether the CAE and the Flourishing Scale are validated and internationally used instruments. For a pediatric oncologists these tools are unfamiliar.

I also wonder the distrbution of the diagnoses of the children. Usually 1/3 is lekemias, 1/3 bran tumors, and the rest all kind of solid tumors. Can you explain?

I also think about the distribution of the patient group otherwise. Why the authors have mixed patients  - would have been better to take only those on therapy + one year post-therapy in order to get more homogenous lifesituation. Or at least make comparisons such that the time interval from diagnosis was taken into the model.

Consentrating on more acute/stressful phhses of treatment might have been able to stronger show the effect/association of parents' resilience (or flourishing) with the well-being.

The current way of pesenting the results is really difficult to understand. In table 2, one maybe can guess that the "x-axis" numbers are pointing to the variables on "y-axis" but it is unclear why the numbering is such as it is, what are the marks (*) depicting, and how one should interprete these. Are the decimal correlation coefficients, and if they are, what does e.g. mean negative correlation coefficient (.251) between  "children and subjective health" or  (.287) between "children and impact caused by stress" or (.654) between "doctors and nurses/nursing assistants"

Please describe clearer what table 2 is describing as the statistical details are not familiar to a clinician in this context,

In figure 1, I cannot find where do come the consepts satisfaction support colleagues, neihbours, or friends, and the consept "treatment status". Thus, it is difficult to check the numbers.

In the discussion, paragraph 3, you should describe better where the reader finds the evidence for your statement tha flourishing was confirmed as a promoter of health and wellbeing? Does it come from table 2 where variable flourishing seems to get most of *-marks?

In the last paragraph of Discussion, I wonder  whether the referencing is correct as there is an interval marked [17-48].

As mentioned above, the authors should mentioned a bit more about the possibilities of training "flourishing", especially as the final sentence in the manuscript suggests such an intervention. One wonders whether such an intervention might be doable or not.

Check references: 2, 17, 20, 41, 44, as some information is missing or is unclear.

Author Response

Response to Reviewer 1

Reviewer 2 Report

The authors measured flourishing and psychosocial functions among parents of children with cancer who are undergoing treatment or have graduated from treatment. They found positive relationships between flourishing and well-being or objective health and proposed that flourishing training can help to improve the coping strategy and wellbeing of parents. 

This is an interesting paper and the results support the conclusion. I am also curious about whether there are "risk factors" of parents with lower flourishing scores and/or worse coping/well-being. Would the socioeconomic factors, cancer type, or treatment status affect the psychosocial functioning and/or flourishing of parents? I suggest the authors to add in such analyses.

Other suggestions: 

Lines 71-77: Can change "it" into "flourishing", so that the readers are easier to read. 

Line 120: Please explain "the framework of the Psychosocial Repercussions o Childhood Cancer in Parents" and cite a reference. 

Page 10, 3rd paragraph, 1st to 2nd lines: There is an colon (:) missing after the word "relationships" and the dash after "health" should be put somewhere else. 

Table 1: Pediatric Brain Tumors may be separated from solid tumors as their disease experience and challenges may be very different. 
What is the definition of "childhood cancer survivor" in Spain? Many study groups define the completion of treatment for 2 years as a survivor. If possible, please further break down the parent numbers of "out of treatment for < 5 years" into "<2 years" and "2-4 years". 

Line 223: Does "this group" mean the entire study population? Please clarify. 

Author Response

Response to Reviewer 2

Round 2

Reviewer 1 Report

Than you for the responses. Please modify the titles of Table 2 and FIgure 1 to inculede a better description of what is presented in thses. I mean that in table 2 you prsente correlations between... and .... Regarding figure 1 you should describe in the title or footneote how the pathanalysis was performed.

Tables and figures should be self-explaining wihout reading the main text.

You also did not reply  (and explain in the text) which patients were included in solid tumors (meaning that whether there were brain tumors included or not).

Author Response

Response to Reviewer 1:

We attach the manuscript again with the "Track changes" function.

We have made the changes you pointed out to us in the last review. Specifically, we have modified the titles of figure 1 and table 2. In this way we think that the information they represent is described. We have also indicated the diagnostic typology at the beginning of the results section.

Thanks for your consideration.

Reviewer 2 Report

The authors have made resonable editing and revision of the manuscript, which is ready for acceptance. 

Author Response

Response to Reviewer 2:

Thanks for your consideration.